# Phytochemical Composition and Biological Properties of *Macleania rupestris* Fruit Extract: Insights into Its Antimicrobial and Antioxidant Activity

**DOI:** 10.3390/antiox14040394

**Published:** 2025-03-27

**Authors:** Arianna Mayorga-Ramos, Johana Zúñiga-Miranda, Elena Coyago-Cruz, Jorge Heredia-Moya, Jéssica Guamán-Bautista, Linda P. Guamán

**Affiliations:** 1Centro de Investigación Biomédica, Facultad de Ciencias de la Salud Eugenio Espejo, Universidad UTE, Quito 170129, Ecuador; arianna.mayorga@ute.edu.ec (A.M.-R.); johana.zuniga@ute.edu.ec (J.Z.-M.); jorgeh.heredia@ute.edu.ec (J.H.-M.); 2Carrera de Ingeniería en Biotecnología de los Recursos Naturales, Universidad Politécnica Salesiana, Sede Quito, Campus El Girón, Av. 12 de Octubre N2422 y Wilson, Quito 170143, Ecuador; ecoyagoc@ups.edu.ec; 3Facultad de Ciencias de la Hospitalidad, Carrera de Gastronomía, Universidad de Cuenca, Cuenca 010201, Ecuador; jessica.guaman@ucuenca.edu.ec

**Keywords:** *Macleania rupestris*, anthocyanins, antimicrobial activity, antibiofilm activity, antifungal activity, in vitro assay

## Abstract

*Macleania rupestris*, a fruit-bearing species of the Ericaceae family, has traditionally been used for its potential medicinal properties. Background/Objectives: This study investigates the phytochemical composition and antimicrobial activity of *M. rupestris* fruit extract, focusing on its antibacterial, antibiofilm, and antifungal effects. Methods: *M. rupestris* (Kunth) A.C.Sm. berries (code: 4456, Herbario QUPS-Ecuador) were collected from the cloud forest Montano Alto, Cuenca-Ecuador, and the extract was obtained using an ethanolic-based extraction and chemically characterized. The antibacterial and antifungal activity of the fruit extract was assessed against seven multidrug-resistant bacteria strains and four fungal strains using the microdilution method. The biofilm inhibition potential was evaluated using a microplate assay with the crystal violet staining method. The antioxidant activity was evaluated using DPPH and ABTS assays. Results: The bioactive compounds showed 853.9 mg phenols/100 g DW, 573.2 mg organic acid/100 g DW, and 21.5 mg C-3-gl/100 g DW of anthocyanins. The antibacterial assays demonstrated significant inhibitory activity against *Enterococcus faecalis*, *Enterococcus faecium*, *Escherichia coli*, and *Staphylococcus epidermidis*, with MIC values ranging from 1.25 to 5 mg/mL. Additionally, the biofilm inhibition assays confirmed the potential of *M. rupestris* extract to disrupt bacterial biofilms, particularly in *S. aureus* and *L. monocytogenes*. Nevertheless, no significant antifungal activity was observed against *Candida* spp., suggesting selective antimicrobial properties. Finally, the antioxidant activity was strong (1.62 mmol TE/100 g DW by DPPH and 3.28 mmol TE/100 g DW by ABTS). Conclusions: These findings indicate that *M. rupestris* possesses promising antibacterial, antibiofilm, and antioxidant properties, which may be attributed to its phenolic and organic acid composition. Further fractionation and targeted bioassays are required to elucidate the specific bioactive compounds responsible for these effects and explore their potential applications in antimicrobial formulations.

## 1. Introduction

*Macleania rupestris*, known in Ecuador as Joyapa, is a fruit-bearing species of the Ericaceae family. It grows in high-altitude regions from Peru to southern Mexico [1]. The plant has a long history of use in traditional medicine, particularly among rural populations. In Ecuador, the *M. rupestris* fruit is consumed for its nutritional value, while its fruits and leaves have traditionally been employed in rural Ecuadorian communities for their possible therapeutic benefits. Its fruit, consumed directly or in juice form, relieves diarrhea and general discomfort [2]. Beverages prepared from its leaves serve as an antidiarrheal remedy and have been used in the treatment of typhoid fever [3], as a tonic [4], to accelerate labor, and to reduce pain [1]. Beverages prepared from its flowers have been used to treat nervous conditions [1]. Early studies have shown that the *M. rupestris* fruit is high in bioactive compounds such as anthocyanins, flavonoids, and polyphenols [3,5,6], contributing to the plant’s antioxidant activity [6,7]. Furthermore, recent research has shown that red raspberry extracts rich in polyphenolic compounds have a good connection with antibacterial activity [8].

Antibiotic-resistant bacterial strains are becoming more prevalent, constituting a serious risk to global health [9]. As conventional antibiotics lose efficiency, there is a growing interest in natural antibacterial antibiotics [10]. Plants, particularly those employed in traditional medicine, represent an attractive opportunity for the development of novel bioactive compounds with antibacterial activity. Despite increasing knowledge of the Ericaceae family’s biological potential [11,12], research on *M. rupestris* is limited [13]. Although previous studies have reported that the ethanolic extracts of its leaves have mild to moderate antibacterial activity against *Escherichia coli*, *Bacillus* spp., and *Staphylococcus aureus* [3], to date, there are no studies that evaluate the biological activity of extracts obtained specifically from its fruits.

This study represents a significant advancement in *M. rupestris* research, as it is the first to evaluate its fruits’ antimicrobial activity. In particular, its antibacterial and antifungal activities against various strains of multi-resistant bacteria and its antibiofilm activity against four bacterial strains were investigated. Additionally, its antioxidant capacity was determined, total polyphenols and anthocyanins were measured, and the extract’s principal polyphenolic components and organic acids were identified using RRLC liquid chromatography. To the best of our knowledge, this is the first study reporting on the antimicrobial activity of the fruit extract of *M. rupestris*. We expect that the findings of this study will help to increase our knowledge of this traditional medicinal plant and contribute to the search for new effective natural compounds with high therapeutic potential.

## 2. Materials and Methods

### 2.1. Plant Material and Physico-Chemical Analysis

*M. rupestris* (Kunth) A.C.Sm. red ripe berries (code: 4456, Herbario QUPS-Ecuador, Appendix A) were collected from the cloud forest Montano Alto, Cuenca-Ecuador (2°53′10.432″ S, 79°5′1.291″ O) (Project MAE-DNB-2019-0911-O). Several approximately 2 kg batches of ripe red fruit were randomly collected in May, the peak production month for the crop in Ecuador. The fruits were collected in refrigerated containers to prevent deterioration during transport. The fruits were washed immediately with plenty of water, and the dried berries were homogenized and divided into two groups. In the first group of 20 berries, the weight, size, pH, soluble solids, titratable acidity, moisture, and ash were quantified [14]. In turn, in a composite sample of 40 additional berries, the ethereal extract (AOAC 922.06), calorific value (FAO), crude fiber (ICC113), total carbohydrate by difference (FAO) and protein (AOAC 2001.11) were also determined in triplicate.

The second portion, which included a composite sample of 2 kg of berries, was frozen at −80 °C and lyophilized in Christ Alpha 1-4 LDplus (GmbH, Osterode am Harz, Germany). The dry powder was ground and stored until analysis [15]. The mineral content of the freeze-dried powder was again quantified in triplicate for Fe, Na, K and Mg. The sample was digested with concentrated nitric acid, and the extract was quantified by atomic absorption spectrometry [14].

### 2.2. Plant Extract

The extraction protocol used in this study was previously described by Barba-Ostria et al. [16], with some modifications. Briefly, the fruit was washed, gridded, and lyophilized to obtain powdered particles. For the ethanolic extraction, 10 g of the powdered material was combined with 200 mL of 96% ethanol. This mixture was then placed in an oil bath and agitated over a magnetic plate stirrer at 70 °C for 60 min.

Subsequently, the resulting solution was filtered using a solid–liquid filtration unit connected to a vacuum pump and the filtrate was collected in a rotary evaporation flask. A total volume of 200 mL of 96% ethanol was added to the solid residue of the initial filtration and mixed until resuspension. This solution was filtered again, and the liquid filtrate was poured into the initial evaporation flask and mixed. The solution was evaporated in a rotary evaporator until the ethanolic solvent was eliminated. Finally, the remaining extract was lyophilized until complete desiccation and the solid extract was weighed.

### 2.3. Phenolic Compound Identification

The quantification of phenolic compounds such as caffeic acid, chlorogenic acid, chrysin, *p*-coumaric acid, *m*-coumaric acid, *o*-coumaric acid, ferulic acid, gallic acid, p-hydroxybenzoic acid, 3-hydroxybenzoic acid, 2-methoxybenzoic acid, 3-methoxybenzoic acid, 3-hydroxybenzoic acid, 2,5-dihydroxybenzoic acid, kaempferol, luteolin, naringin, quercetin, rutin, shikimic acid, syringic acid, and vanillic acid was carried out according to the method described by Coyago et al. [17]. The sample was extracted with an 80% methanol solution, acidified with 0.1% HCl, and then shaken under ultrasound. The extract was quantified on an RRLC 1200 liquid chromatograph (Agilent Technologies, Santa Clara, CA, USA) coupled to a DAD-UV-Vis detector at a wavelength between 220 and 500 nm, using a Zorbax Eclipse Plus C18 column (4.6 × 150 mm, 5 um) (Agilent Technologies, Santa Clara, CA, USA). The mobile phase consisted of 0.01% formic acid in water (solvent A) and acetonitrile (solvent B), with a linear gradient elution as follows: 100% A at 0 min; 95% A + 5% B at 5 min; 50% A + 50% B at 20 min; and washing and re-balancing of the column at 22 min. Each phenolic compound was expressed in grams per 100 g dry weight (g/100 g DW). Total phenolic compounds were calculated by summing the concentrations of the individual compounds. Analysis was performed in triplicate.

### 2.4. Organic Acid Identification

The quantification of organic acids (citric, malic and tartaric acid) was performed according to the protocol of Coyago et al. [15]. Organic acids were extracted from the lyophilized powder using a sulphuric acid solution and ultrasonic agitation. The extract was quantified on an RRLC 1200 liquid chromatograph (Agilent Technologies, Santa Clara, CA, USA) coupled to a DAD-UV-VIS detector at 210 nm and a YMC-Triart C18 column (150 × 4.6 mm column size, 3 µm particle size, 12 nm pore size, 400 bar pressure limit) (YMC Europe GmbH, Dinslaken, Germany). The mobile phase consisted of 0.027% sulphuric acid, and the total run time was 30 min. Each organic acid was expressed as grams per 100 g dry weight (g/100 g DW). Total organic acids were calculated by summing the concentrations of the individual compounds. Analysis was performed in triplicate.

### 2.5. Total Anthocyanins

For the extraction of anthocyanins, 40 mg of lyophilized powder was mixed with 2 mL of ethanol. The mixture was homogenized and vortexed under ultrasound for 3 min. The supernatant was separated by centrifugation at 1400 rpm for 5 min at 4 °C. Then, 50 µL of the supernatant was mixed with 200 µL of a solution of 0.025 M potassium chloride buffer, pH 1, and 50 µL of the supernatant was mixed with 200 µL of 0.4 M sodium acetate buffer, pH 4.5. The absorbance of the two solutions was measured at 520 nm and 700 nm in a spectrophotometer with a microplate reader. The calibration curve was established using a cyanidin-3-glucoside chloride standard in the concentration range 0.05 to 0.2 mg/mL. The concentration was expressed as mg cyanidin-3-glucoside chloride per 100 g dry weight (mg C-3-gl/100 g DW). Analysis was performed in triplicate.

### 2.6. Antibacterial Activity Assay

The antibacterial activity of the ethanolic extract of *M. rupestris* was assessed against seven multidrug-resistant bacteria, *Klebsiella pneumoniae*, *Escherichia coli*, *Salmonella enterica* serovar *Kentucky*, *Enterococcus faecalis*, *Staphylococcus epidermidis*, *Enterococcus faecium*, and *Pseudomonas aeruginosa,* using the microdilution method [18]. These seven clinical multi-drug-resistant isolates were provided by the National Health Institute of Ecuador (INSPI), and their resistance profiles can be found in Appendix A.

The bacterial inoculum was prepared in a brain-heart infusion broth (BHI) to a final cell density of 5 × 10^5^ CFU/mL. Stock solutions of the tested compound were prepared by dissolving them in dimethyl sulfóxido (DMSO) at 0.8 g/mL. Tested volumes were adjusted so the final concentration of the DMSO in each well was always 2.5% *v*/*v*. This concentration was previously shown not to affect bacterial growth [19]. Bacterial cells were grown with 2.5% DMSO as a control to rule out any potential growth inhibitory effect. Additionally, nourseothricin (100 µg/mL) was used as a control for growth inhibition. BHI alone, or BHI supplemented with the extract at different concentrations, were used as blanks.

The extract sensitivities were assessed via the microdilution method and according to the Clinical and Laboratory Standards Institute (CSLI) guidelines [20], with the following modifications: First, from the stock solution, the extract was serially diluted in DMSO, then 5 µL of each dilution was added to 195 µL of bacterial suspension (5 × 10^5^ cfu/mL) to a total volume of 200 µL. The plates were then incubated at 37 °C for 20 h with constant shaking at 300 cpm (double orbital setting). The minimal inhibitory concentration (MIC) was determined by comparing the OD_600_ at time 0 with the value after 24 h in samples exposed to the tested suspension. The MIC was defined as the lowest concentration of the antibacterial agent, which completely inhibited the growth of the microorganism as determined by the optical density at 600 nm. These assays were performed at least in triplicate.

### 2.7. Antifungal Activity Assay

The minimal inhibitory concentration (MIC) of the extract against four *Candida* strains (*Candida krusei* ATCC 14243, *Candida albicans* ATCC 10231, *Candida glabrata* ATCC 66032, *Candida tropicalis* ATCC 13803) was assessed using the microdilution method, following the Clinical and Laboratory Standards Institute (CLSI) guidelines with modifications [21,22]. The extract was serially diluted in distilled water (ranging from 100 to 1.25 μg/mL), and 10 μL of each dilution was combined with 190 μL of a fungal suspension (5 × 10^5^ CFU/mL) to achieve a final volume of 200 μL per well. Plates were incubated at 37 °C for 72 h with constant shaking (200 rpm), and the OD_600_ was measured immediately after inoculation (time 0) and after 72 h using a Cytation5 multi-mode plate reader (BioTek). The MIC was determined by comparing the OD_600_ at time 0 with the value after 72 h for samples exposed to different concentrations of the test suspension alongside positive (nourseothricin, 100 µg/mL) and negative (H_2_O) controls. The MIC was defined as the lowest concentration of the extract that completely inhibited microorganism growth, as indicated by the optical density at 600 nm. All assays were conducted in triplicate.

### 2.8. Antibiofilm Activity

The inhibitory effect of the *M. rupestris* extract on biofilm formation of four bacterial strains (*S. aureus* ATCC 25923, *L. monocytogenes* ATCC 13932, *P. aeruginosa* ATCC 9027 and *Burkholderia cepacia* ATCC 25416) was investigated using the microplate assay with crystal violet staining, as described by Merritt et al. [23], with some modifications. Briefly, the bacterial strains were grown in TSB+G (Tryptic Soy Broth medium supplemented with 1% Glucose) overnight at 37 °C. The next day, 1:100 bacterial dilutions were prepared from the overnight cultures and evaluated against a range of fruit extracts (30 to 0.1 mg/mL). Then, 150 μL aliquots were added to 96-polystyrene plates and incubated at 37 °C in static conditions for 24 h. The medium containing free-floating bacteria was carefully removed using a micropipette, and the wells were washed twice with PBS buffer 1x (pH 7.2). The plate was dried for 1 h at 60 °C inside a laboratory oven. Subsequently, the biofilms were stained with 150 μL of 0.1% crystal violet for 20 min at room temperature and washed three times with PBS buffer 1x (pH 7.2). Finally, 150 μL of 96% ethanol was added to each well for 30 min, and the absorbance was measured at 590 nm using a plate reader Cytation 5 System (Agilent Biotek, Winooski, VT, USA). Each dilution assessment was performed in three technical replicates, and the experiment was performed in three individual biological trials. Each plate included a positive control of three technical replicates by diluting 1:100 of the bacterial overnight culture in TSB+G.

The biofilm inhibition percentage was assessed with the following formula:Inhibitory rate (%) = [(Positive control OD_590_ nm − Sample OD_590_nm)/Positive control OD_590_ nm] × 100

### 2.9. Antioxidant Activity

Antioxidant activity was measured using the DPPH (2,2-diphenyl-1-picrylhydrazyl) and ABTS (2,2′-Azino-bis(3-ethylbenzothiazoline-6-sulfonic acid)) method described by Coyago et al. [24]. For the DPPH method, an extract was prepared by mixing 20 mg of lyophilized powder with 2 mL of HPLC-grade methanol. The mixture was stirred in an ultrasonic bath (Fisher Scientific Inc., Waltham, MA, USA), and the supernatant was collected for quantification. For the ABTS method, an extract was prepared by mixing 20 mg lyophilized powder with 400 µL methanol and 400 µL distilled water in an ultrasonic bath. The supernatant was then recovered by centrifugation, and the solid was re-extracted with 560 µL acetone and 240 µL distilled water. The extracts were assayed with ABTS•+ or DPPH- radical, and the absorbance was quantified using a microplate reader spectrophotometer (Agilent Scientific Instruments, Santa Clara, CA, USA). A trolox standard was used, and the concentration was expressed as mmol Trolox equivalents per 100 g dry weight (mmol TE/100 g DW).

### 2.10. Statistical Analysis

Statistical analysis for the Biofilm Inhibition Evaluation was performed using a two-way ANOVA Dunnett test to determine the significance of the differences across groups using the GraphPad Prism 10.2 software (GraphPad Software Corp, San Diego, CA, USA). The *p* values < 0.05, <0.01, and <0.001 were considered statistically significant.

## 3. Results and Discussion

### 3.1. Physico-Chemical Analysis

Table 1 shows the average values of the physico-chemical parameters of *M. rupestris* berries. This species is a plant native to the Andes and is valued for its edible fruit and potential health benefits [25]. The fruit of *M. rupestris* is a berry that provides a limited edible portion (2.1 g) and has an almost spherical or slightly elliptical morphology, with an equatorial diameter of 14.3 mm and a longitudinal diameter of 14.5 mm.

The acid pH of the fruit (3.7) is similar to that of fruits such as citrus fruits, whose pH ranges from 2.9 to 5.8 [26], suggesting a moderate acidity that may contribute to both its microbiological stability and its flavor. Despite this acidity, the soluble solids content is considerable (10.3 °Brix), indicating the presence of water-soluble compounds such as vitamins and polyphenols, which are responsible for a sweetness similar to that of some tropical fruits [15]. This sweetness is in balance with its low titratable acidity (0.8%), which suggests a harmonious combination of acidity and sugar, a characteristic that enhances its sensory appeal.

In terms of moisture content, the fruit has a high amount (82.6%), which indicates its highly perishable nature and the need for appropriate conservation conditions to prevent microbial deterioration [27]. On the other hand, the ash content (0.2%) reflects a low concentration of total minerals, indicating a limited mineral supply.

*M. rupestris* is a species with low levels of lipids and protein, which is common in plant products with low caloric density (69 kcal) and high fiber content (4.3 g/100 g). At the same time, the total carbohydrate intake suggests that this fruit is likely to be composed mainly of simple sugars. This makes *M. rupestris* a fruit with nutritional potential, as dietary fiber is important in regulating intestinal transit and may have beneficial effects on glycaemic control and cholesterol reduction, as well as contributing to a feeling of satiety.

In terms of macroelement content in *M. rupestris*, magnesium was found to have the highest concentration in the fruit, followed by potassium. Potassium plays a crucial role in plant physiology, influencing cell growth, nutrient transport, and stress response [28]. Magnesium is essential for several physiological processes in humans, and its deficiency is associated with an increased risk of developing chronic diseases [29]. Although no specific data on the mineral content of *M. rupestris* have been reported to date, the values found for magnesium and potassium were higher than those reported for spinach (55 mg/100 g dry weight for magnesium and 539 mg/100 g dry weight for potassium). This suggests that the consumption of this fruit could contribute significantly to meeting the minimum daily requirements for potassium (3500 mg/day), magnesium (300 mg/day), calcium (950 mg/day), and even iron, positioning it as a potential dietary alternative [30].

Table 2 shows the average values of the phenolics and organic acids of *M. rupestris*, while Figure 1 shows the resulting chromatogram of the phenol profile. This berry displayed a high concentration of phenolic acids belonging to the benzoic acid group, such as gallic acid, syringic acid, and *p*-coumaric acid. Flavonoids, such as quercetin, and flavanols, such as catechin, known for their antioxidant, anti-inflammatory, and antimicrobial properties, were also identified [7,31]. Thus, fruit phenolic concentrations can vary depending on the geographical location of the crop, the agronomic conditions, and the environmental conditions, as shown in previous studies [17,32].

The total phenolic concentration, calculated as the sum of the individual compounds, was 853.9 mg/100 g dry weight (DW). However, this amount was lower than that reported by other studies, which reported concentrations of 1704.6 mg GAE/100 g extract [7]. These discrepancies may be due to differences in the analytical methods used. Chromatographic methods generally offer higher precision and sensitivity, allowing compound specific separation and identification. On the other hand, spectrophotometric methods, such as the Folin–Ciocalteu reagent, may react with other reducing substances present in the extract, which could lead to an overestimation of total phenolics [33].

In turn, *M. rupestris* showed high concentrations of tartaric acid, which is a rare acid in fruits but contributes to the sour taste, as reported in other studies [15]. The total anthocyanin content observed in this study was consistent with the values reported for phenolic compounds. However, the concentrations obtained were lower than those described by other authors, who reported a concentration of 1467.20 mg cyanidin-3-glucoside/100 g in an acidified ethanolic extract obtained from the fruit that had been oven-dried at 60 °C for 52 h and extracted by shaking at 70 °C for 1 h [7].

### 3.2. Minimum Inhibitory Concentration of Multidrug-Resistant Bacteria

The determination of the minimum inhibitory concentration (MIC) is fundamental in monitoring resistance development and establishing optimal pharmacodynamic dosing. The antimicrobial activity of *M. rupestris* extract was evaluated against seven clinically significant microbial strains.

The in vitro susceptibility tests indicated that *M. rupestris* extract exhibited significant inhibitory effects against Gram-negative bacteria, *E. coli*, and Gram-positive bacteria, *S. epidermidis,* with an MIC value of 5 mg/mL. However, the lowest MIC was observed against Gram-positive bacteria, *E. faecalis* and *E. faecium*, at 1.25 mg/mL. In contrast, the *M. rupestris* extract did not work against *K. pneumoniae*, *S. enterica* serovar *Kentucky*, and *P. aeruginosa*. Table 3 presents the MIC for each microbial strain as determined by the microdilution method.

The antibacterial assessment of the *M. rupestris* fruit reveals promising inhibitory effects against several bacterial strains, underscoring its potential as an antibacterial agent. The bioactivity of *M. rupestris* fruit extract could be attributed to its phenolic content, particularly to quercetin and *p*-coumaric acid, which are known for their antibacterial properties [34]. *p*-Coumaric acid has demonstrated broad-spectrum antibacterial activity; for example, Elansary et al. reported that *p*-coumaric acid present in *Ruta graveolens* was effective against various bacterial species, such as *E. coli* [34]. Quercetin, known for its efficacy against both Gram-positive and Gram-negative bacteria, including methicillin-resistant *S. aureus* (MRSA) and *S. epidermidis* [35], could contribute to the inhibition observed in the bacterial strains within this study [34]. The antibacterial mechanisms of these phenolic compounds likely involve disrupting bacterial cell membranes, inhibiting essential enzymatic functions, and interfering with metabolic pathways, thus impairing bacterial growth [36].

In addition to phenolics, the organic acid profile of *M. rupestris* reveals a total organic acid content of 573.2 mg/100 g DW, with tartaric acid being the most prevalent, followed by malic acid. Organic acids are recognized for their ability to lower pH and create an inhospitable environment for bacteria, thereby enhancing the antibacterial activity of plant extracts [37]. Another mechanism could be the depression of the microbial cell’s internal pH by ionizing undissociated acid molecules or disrupting substrate transport by altering cell membrane permeability. In addition, Gram-positive bacteria have a simpler structure, with a thick layer of peptidoglycan and a single lipid bilayer [38]. All these possible mechanisms found in the literature could explain their greater sensitivity to organic acids of the strains tested in this study, *E. faecalis* and *E. faecium*.

Gram-negative bacteria usually display more complex resistance mechanisms than Gram-positive bacteria; the hydrophobic external membranes in Gram-negative bacteria can hinder the entry of low molecular weight hydrophilic molecules, as they can only cross the membrane through water channels formed by transmembrane proteins, porins, in the lipid bilayer [39]. This could explain why our Gram-negative bacteria, such as *K. pneumoniae*, *S. enterica* serovar *Kentucky*, and *P. aeruginosa*, did not show sensitivity to the extract. However, the extract had an effect against *E. coli* despite the bacteria being Gram-negative. The literature suggests that some bacteria, such as *E. coli*, have various efflux pumps, but their activity can vary, and not all pumps are equally effective against every compound [40]. Based on this information, we hypothesized that the *E. coli* bacterial strain tested in this study could have less active efflux pumps to the extract, preventing it from being expelled from the interior of the bacterial cell and thus increasing its effectiveness. Our results suggest that the antibacterial activity of the compounds found in the fruit extract is limited to certain bacterial species.

### 3.3. Evaluation of Anti-Fungal Activity

The antifungal activity of *M. rupestris* extract was tested against four *Candida* species (*C. albicans, C. glabrata*, *C. krusei*, and *C. tropicalis*) at extract concentrations of 0.5 to 20 mg/mL. The findings revealed no inhibition of fungal growth in any of the tested species, regardless of the extract concentration (Appendix A). The antifungal assay results indicated that *M. rupestris* extract did not exhibit inhibitory activity against any *Candida* species tested, even at high concentrations. This outcome contrasts with the antibacterial and antibiofilm activity observed in the study, suggesting differences in the mechanisms of action of the compounds present in the extract against bacteria and fungi.

The lack of antifungal activity in the *M. rupestris* extract can be explained by several factors. First, while the *M. rupestris* extract is rich in phenolic compounds, the antibacterial action observed against *E. coli*, *S. epidermidis*, *E. faecalis*, *and E. faecium* has been attributed to the ability of these compounds to alter cell membrane integrity, inhibit key enzymatic systems, and modulate quorum sensing signaling [41]. However, fungi have more complex cellular structures, with a cell wall rich in chitin and β-glucans, which could limit the ability of these compounds to exert an inhibitory effect [42,43].

Second, anthocyanins have demonstrated antioxidant potential and some antimicrobial activity in other contexts [44,45]. However, their antifungal activity often depends on interactions with fungal lipid membranes, and their hydrophilic nature may limit such interactions, reducing their efficacy. Additionally, previous studies have indicated that the antifungal efficacy of anthocyanins tends to be more pronounced in extracts enriched with lipophilic derivatives or in synergy with other bioactive compounds [46,47].

Third, another relevant factor is the presence of carbohydrates and fiber in *M. rupestris*. The high concentration of sugars and water-soluble compounds may have contrasting effects on the extract’s bioactivity [48]. While in bacteria, these compounds may contribute to osmotic disruption, in fungi, it can promote growth by providing easily metabolizable carbon sources.

Fourth, another plausible reason is the insufficient concentration of active compounds within the extract, which may fall below the threshold required for effective antifungal action. Although quercetin and other flavonoids are well recognized for their antifungal potential, their efficacy in extracts is often diminished due to dilution, reducing their overall potency against fungal cells [49]. Furthermore, interactions among the various constituents of the extract could play a crucial role in modulating its antifungal activity.

Finally, *Candida* species are known for their metabolic plasticity, allowing them to utilize a wide range of carbon sources, including polyphenols and organic acids, as nutrients rather than encountering them as inhibitory agents [50]. This could explain why the extract, despite its high phenolic content, did not exhibit antifungal effects. Future studies should consider fractionating the extract to isolate potentially active antifungal compounds and test their efficacy individually or in combination with established antifungal agents.

### 3.4. Evaluation of Biofilm Inhibition Activity

The extract was tested to assess the biofilm inhibition activity against biofilm-forming microorganisms such as *S. aureus*, *L. monocytogenes*, *P. aeruginosa*, and *B. cepacia*. The MBIC_50_ (minimum biofilm-inhibiting concentration for 50% inhibition effectiveness) was calculated and shown in Figure 2. The inhibition rates of all the tested concentrations can be seen in Appendix A. The extract displayed inhibitory concentration in all the strains used during this evaluation, and the lowest BMIC_50_ value was observed in three bacterial strains. *S. aureus* was significantly inhibited at 1 mg/mL (71 ± 3.31% inhibition), *L. monocytogenes* was inhibited at 1 mg/mL (72 ± 8.63% inhibition), and *B. cepacia* recorded the MBIC at 1 mg/mL (69 ± 7.69% inhibition). *P. aeruginosa* required a higher extract concentration for the MBIC_50_ at 10 mg/mL (76 ± 2.91% inhibition).

The utilization of naturally occurring plant compounds has shown promise in preventing and treating infections, particularly in targeting biofilm formation by pathogenic bacteria. Our study has identified anti-biofilm activity in all four bacterial strains. Upon inspection of the chemical composition of *M. rupestris*, some compounds which have already been correlated with strong anti-biofilm activities were identified.

Derived from a variety of foods, the literature suggests that compounds like quercetin disrupts biofilm establishment in several of the strains analyzed during our study [51]. For instance, some studies have shown that quercetin prevents *L. monocytogenes* colonization on stainless steel, reduces extracellular protein levels in biofilms, and suppresses bacterial adhesion genes [52]. This adhesion inhibition capability is crucial as it can impede the biofilm formation of *Listeria* spp. over food processing surfaces, reducing the risk of foodborne contamination [53,54]. Similarly, In *S. aureus*, quercetin demonstrates an equally powerful impact by decreasing biofilm production and suppressing the expression of genes linked to biofilm formation, such as *icaA* and *icaD*, and quorum-sensing genes, including *agrA* [55]. Additionally, in *P. aeruginosa*, which affects immunocompromised patients and plays a critical role in cystic fibrosis outcomes, quercetin impedes biofilm formation and motility [56]. Studies show that quercetin possesses the ability to block the las and *rhl* quorum-sensing pathways, which are essential in biofilm development, offering a potential alternative to conventional antibiotics [56]. Its effectiveness, in some cases, even surpasses azithromycin, a commonly used antibiotic [57]. Thus, quercetin provides a multi-faceted approach in combating biofilm-associated infections, offering promising insights into natural compounds as adjunctive therapies.

Moreover, our analysis observed an important presence of *p*-coumaric acid, which has been poised in the literature as a potent anti-microbial agent [58]. Some studies have observed the presence of *p*-coumaric acid in natural extracts from natural sources like burdock leaves and demonstrated a strong inhibition of *P. aeruginosa* biofilm formation [59]. A recent study examined the effects of *p*-coumaric acid in *L. monocytogenes*, and it was found that *p*-coumaric acid was a potent inhibitor of the gene *RecA*, related to repair and maintenance of bacterial DNA [60]. The mechanism underlying such inhibitory action of *p*-coumaric acid involves its ability to interfere with the DNA binding domain of the RecA protein [60]. *p*-Coumaric acid also potentiates the activity of ciprofloxacin by inhibiting the drastic cell survival of *L. monocytogenes*, as well as the filamentation process, which is a defensive mechanism in response to DNA damage [60]. Other studies have also assessed the synergizing effect of *p*-coumaric acid with existing antibiotics with some promising results that may suggest that *p*-coumaric acid could potentially enhance the shelf life of antibiotics and aid with the emergence of antibiotic resistance in bacteria [61]. Furthermore, other studies have also shown that gallic acid and *p*-coumaric acid have the ability to block expression of flagella gene (*flgA*) in *Pseudomonas fluorescens*, significantly decreasing the rate of microorganism colonization on stainless-steel surfaces [62]. Overall, a range of the phenolic compounds found in our *M. rupestris* extract have been linked to the relevant biofilm inhibition properties.

Tartaric acid and malic acid were identified as the primary organic acids in our extract, both of which have been previously evaluated in the literature for their antimicrobial and antibiofilm properties. Organic acids are widely utilized as food preservatives due to their inherent antimicrobial potential [63]. Their mechanism of action involves the permeation of undissociated molecules into microbial cell membranes, where they dissociate and lower intracellular pH, thereby inhibiting microbial growth [64]. The antimicrobial efficacy of organic acids has also been attributed to their ability to decrease gastric pH, creating unfavorable conditions for pathogenic bacteria [65]. For example, malic acid has been shown to effectively inhibit *Salmonella enterica* serovar *Typhimurium* biofilm formation on carrots and other food-contact surfaces [66]. Furthermore, other studies have demonstrated that citric acid, malic acid, and tartaric acid exhibit significant antibacterial activity against pathogens such as *E. coli* O157:H7, *L. monocytogenes*, and *Salmonella Typhimurium* [67]. In food processing, the novel derivatives of tartaric acid have exhibited high antimicrobial activity against multidrug-resistant bacteria while maintaining biodegradability through soil microflora, suggesting their potential application in food packaging as safe antimicrobial alternatives [68]. Recent research has also explored strategies to potentiate the antimicrobial effects of tartaric acid, including its combination with plasma-activated water (PAW) [69]. A 2023 study demonstrated that the combined treatment of PAW and tartaric acid induced oxidative damage to the *S. aureus* cell membrane, compromising its defense mechanisms [69]. This approach resulted in a reduction in *S. aureus* cells in a fresh-cut pineapple model at different temperatures, which suggests that it can increase the shelf life of the fruit [69]. Further studies concerning our *M. rupestris* extract are recommended to unveil the principal compounds responsible for its strong biofilm inhibition activity and the mechanisms underlying such activity.

### 3.5. Evaluation of Antioxidant Activity

The antioxidant activity of the *M. rupestris* extract was evaluated by the DPPH and ABTS assays, as shown in Table 4. According to the DPPH assay, the *M. rupestris* extract had a value of 100.1 μg/mL which is categorized as strong antioxidant activity. According to [70,71], the antioxidant activity is considered very powerful for IC_50_ values lower than 50 μg/mL, strong for values between 50 and 100 μg/mL, moderate for values between 101 and 250, weak for values that fall in the range between 250 and 500 μg/mL and inactive for values higher than 500 μg/mL. The antioxidant activity found by this study was higher than that reported by [6]. This difference might be attributed to the solvent used for extraction. For instance, one of the first studies concerning this plant prepared a *M. rupestris* aqueous extract and obtained an IC_50_ value of 29,240 μg/mL using the DPPH assay [6]. It has been found that for chilli varieties, a DMSO extract shows, significantly, the highest value of *in vitro* antioxidant activity compared with the other common solvents used [72].

The antioxidant activities of other members of the Ericaceae family have been reported by [73]. They studied dry extracts from five species—*Arbutus unedo*, *Bruckentalia spiculifolia*, *Calluna vulgaris*, *Erica arborea* and *Erica carnea*—and found that all of them showed very powerful antioxidant activity. *Arbutus unedo* presents the highest radical scavenging activity (IC_50_ 7.17 ± 0.46 μg/mL). Another study examined the antioxidant activity of another member of the Ericaceae family, *Vaccinium uliginosum* L. They found an IC_50_ value of 85.8 μg/mL for the crude extract measured by the DPPH assay [74].

The strong antioxidant activity of the *M. rupestris* extract could be attributed to phenolic acids such as p-coumaric acid, catechin, and gallic acid, which have been identified (Table 2). Indeed, phenolic compounds contribute mainly to the scavenging activity of free radicals [75]. In addition, flavonols [76] like quercetin-hexoside and quercetin-pentoside also contribute to antioxidant activity. Notably, the highest concentration macroelement was found to be magnesium (576.5 mg/100 g DW Table 1), which is an antioxidant [77]. It might form complexes with organic acids (tartaric or malic) or other components. Indeed, it has been reported that quercetin’s free radical scavenging activity increased after the complexation of magnesium (Mg^+2^) cation [78,79]. In 2015, a study examined a metal–flavonoid complex based on analytical and spectral data and found that the complex showed higher antioxidant activity than quercetin alone [78]. The flavonoid antioxidant activity was linked to the amount and positions of the OH groups in its structure [78]. This indicates that the metal ion alters quercetin’s chemistry, boosting its antioxidant activity. Therefore, *M. rupestris* could be a great source of this mineral to contribute to the appropriate homeostasis in the body as it mitigates the effects of oxidative stress [80].

According to the ABTS test, the *M. rupestris* extract had a value of 215.24 μg/mL. The results of the antioxidant activity using the ABTS method were higher than the results using DPPH. These differences among antioxidant assays might be attributed to their characteristics. For instance, the polarity of antioxidants have reported that the DPPH assay is suitable for hydrophobic solvents (only organic solvents) like DMSO, while the ABTS assay works well with hydrophilic and lipophilic solvents [81]. Furthermore, it is possible that the incubation time in the ABTS assay was insufficient to achieve the maximum decolorization of the ABTS radical solution. Although the reaction mechanism of the DPPH and ABTS assays are the same (mixed-mode hydrogen atom transfer and single-electron transfer), the incubation time of ABTS assay is shorter than the incubation time of the DPPH assay, as the initial electron transfer is much faster in the ABTS assay due to sterically hindered DPPH radical site, which is complex for phenols to access [82]. In addition, a low correlation between the DPPH and ABTS assays has been reported [81,83]. Notably, antioxidant compounds can respond in a different manner to different radical or oxidant compounds [84].

## 4. Conclusions

*M. rupestris* berries are notable for their interesting physicochemical profile, which combines acidity and sweetness, making them sensorially pleasant. Their high moisture and fiber content, low fat and protein content, and high levels of magnesium and potassium suggest interesting nutritional potential, like in the regulation of intestinal transit and glycemic control. Although their high perishability requires proper storage, the bioactive compounds in these berries, such as flavonoids and phenolic acids, provide significant nutritional and health benefits. The total polyphenol and total anthocyanin content of *M. rupestris* extract was found to be lower than those reported in the previous studies; however, this extract has remarkable antibacterial potential, particularly against strains of *E. faecalis* and *E. faecium* (MIC = 1.25 mg/mL) and, to a lesser extent, against *E. coli* and *S. epidermidis* (MIC = 5 mg/mL), but no antifungal activity was observed. It is worth noting that this extract has remarkable biofilm inhibitory activity against various pathogenic bacteria, with efficacy at concentrations as low as 1 mg/mL. Furthermore, this extract has strong antioxidant activity (1.62 mmol TE/100 g DW by DPPH and 3.28 mmol TE/100 g DW by ABTS assay). In conclusion, this study shows that the extract obtained from *M. rupestris* showcases strong potential as a source of bioactive compounds with health and nutrition applications. The novelty of this research lies in its focus on the fruit rather than the more extensively studied leaves, which contributes valuable new insights. However, limitations such as the lack of in vivo validation and unexplored synergistic effects with existing antibiotics should be addressed in future studies. Despite the positive results in the biological activities studied, further research is required to uncover the specific mechanisms of action and potential therapeutic applications.

## Figures and Tables

**Figure 1 antioxidants-14-00394-f001:**
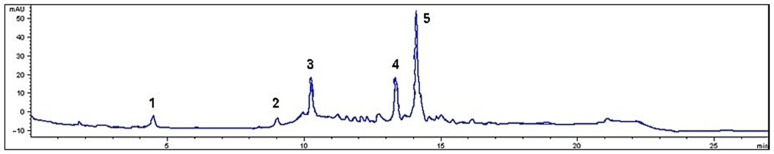
Chromatogram of *M. rupestris* phenolics at 280 nm. The numbers on the peaks of the chromatogram indicate the individual phenolic compounds: 1. Gallic acid; 2. Catechin; 3. Syringic acid; 4. Quercetin; 5. *p*-Coumaric acid.

**Figure 2 antioxidants-14-00394-f002:**
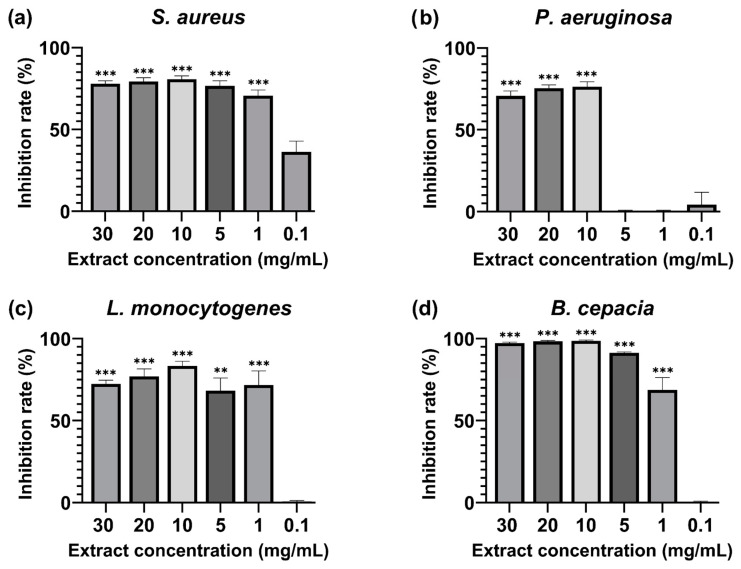
Percentage of 50% biofilm inhibition of (**a**) *Staphylococcus aureus* ATCC 25923, (**b**) *Pseudomonas aeruginosa* ATCC 9027, (**c**) *Listeria monocytogenes* ATCC 13,932, and (**d**) *Burkholderia cepacia* ATCC 25,416 after 24 h incubation with *M. rupestris* extract at a 30–0.1 mg/mL concentration. Treatments at different concentrations were compared with a 50% theoretical inhibition standard for statistical significance using a two-way ANOVA test. All the values are mean ± SD, *p*-value (**) < 0.01, (***) < 0.001.

**Table 1 antioxidants-14-00394-t001:** Average values of the commercial quality parameters of *M. rupestris*.

Parameters	Value
Weight (g)	2.1 ± 0.4
Equatorial diameter (mm)	14.3 ± 1.4
Longitudinal diameter (mm)	14.5 ± 1.2
pH	3.7 ± 0.6
Soluble solids (°Brix)	10.3 ± 0.6
Total titratable acidity (%)	0.8 ± 0.1
Moisture (%)	82.6 ± 1.3
Ethereal extract (g/100 g) *	0.7 ± 0.0
Calorific value (kcal) *	69.0 ± 0.1
Ash (%) (200 mg/100 g)	0.2 ± 0.0
Fe (mg/100 g DW ^1^) *	8.4 ± 0.6
Na (mg/100 g DW ^1^) *	74.5 ± 9.4
K (mg/100 g DW ^1^) *	241.6 ± 54.7
Mg (mg/100 g DW ^1^) *	576.5 ± 6.1
Fibre (g/100 g) *	4.3 ± 0.0
Total carbohydrate (g/100 g) *	15.1 ± 0.2
Protein (g/100 g)	1.0 ± 0.1

^1^ DW: Dry weight. The mean is based on 20 scores. * The mean is based on 6 scores.

**Table 2 antioxidants-14-00394-t002:** Average values of phenolics and organic acids of *M. rupestris*.

Parameters	Value		
Phenolics (mg/100 g DW ^1^)			
Gallic acid	14.9	±	0.0
Catechin	18.5	±	5.9
*p*-Coumaric acid	511.1	±	11.0
Syringic acid	151.9	±	6.0
Quercetin	157.5	±	7.2
Total phenolics	853.9	±	18.3
Organic acids (mg/100 g DW ^1^)			
Malic acid	189.4	±	18.5
Tartaric acid	306.0	±	13.4
Citric acid	77.8	±	4.6
Total organic acid	573.2	±	36.6
Total anthocyanins (mg C-3-gl/100 g DW ^1^)	21.5	±	2.5

^1^ DW: Dry weight. The mean is based on 6 scores.

**Table 3 antioxidants-14-00394-t003:** Minimal inhibitory concentration by growth kinetics over 20 h (OD_600_) after serial microdilution in 96-well plates.

Bacteria Strain	MIC (mg/mL)
*K. pneumoniae*	NA ^1^
*E. coli*	5.00
*E. faecalis*	1.25
*S. epidermidis*	5.00
*E. faecium*	1.25
*S. enterica serovariedad Kentucky*	NA ^1^
*P. aeruginosa*	NA ^1^

^1^ NA: Without activity at tested concentrations.

**Table 4 antioxidants-14-00394-t004:** Antioxidant activity measured by DPPH and ABTS methods.

Compound	DPPH (mmol TE/100 g DW)	ABTS mmol TE/100 g (DW)
*Macleania rupestris*	1.62 ± 0.09	3.28 ± 0.06
IC_50_ Trolox (mM)	0.23 ± 0.04	0.43 ± 0.02

## Data Availability

The original data presented in the study are openly available in FigShare at DOI: 10.6084/m9.figshare.28489226.

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
