# Peer review of "Phytochemical Composition and Biological Properties of *Macleania rupestris* Fruit Extract: Insights into Its Antimicrobial and Antioxidant Activity"

_antioxidants, 2025, doi:10.3390/antiox14040394_

Round 1

Reviewer 1 Report

This paper is well-structured and contributes to the field of natural antimicrobial agents. With minor refinements in the discussion (particularly study limitations) and a more explicit connection to prior research on M. rupestris, it would be suitable for publication.

The title, "Phytochemical Composition and Antimicrobial Properties of Macleania rupestris Fruit Extract: Insights into Its Antibacterial, Antibiofilm, and Antifungal Activity", accurately describes the paper’s content. It is specific and informative, clearly indicating the study's focus on the phytochemical composition and antimicrobial effects of Macleania rupestris fruit extract.

The introduction provides a well-rounded overview of Macleania rupestris, covering its traditional medicinal uses, phytochemical composition, and the relevance of studying its antimicrobial properties. The background on antibiotic resistance and the need for alternative natural compounds is well-integrated. However, while the introduction references prior research on related species, it could provide a more in-depth discussion of previous studies specifically on M. rupestris fruit.

The research design appears appropriate for evaluating the antimicrobial potential of M. rupestris fruit extract. The methods for phytochemical characterization, antibacterial and antifungal assays, and biofilm inhibition tests are described in detail. The use of standard protocols, such as the microdilution method and RRLC liquid chromatography, strengthens the study’s credibility. However, additional clarification on controls used in antimicrobial assays and potential extraction limitations would improve transparency.

The results are presented in a clear and structured manner, with data tables and figures effectively illustrating findings. The conclusions are well-supported by the results, emphasizing the selective antibacterial activity of the extract and its biofilm inhibition potential. The discussion effectively places findings in the context of existing literature, explaining potential mechanisms of action. However, the lack of antifungal activity is briefly addressed and could benefit from a more detailed discussion.

The cited references are relevant and up-to-date, covering phytochemistry, antimicrobial mechanisms, and prior research on natural antibacterial compounds. There is a good balance of primary research articles and reviews. However, a more extensive discussion of studies related to M. rupestris fruit extract would strengthen the literature review.

This study provides a valuable contribution by identifying potential antibacterial and antibiofilm properties of M. rupestris, particularly against drug-resistant bacteria. The paper’s novelty lies in its focus on the fruit extract rather than the leaves, which have been more extensively studied. Further studies on compound fractionation and mechanisms of action would enhance the contribution.

The discussion does not explicitly address study limitations, which should be included. Potential limitations include:
   The study does not explore the synergistic effects of M. rupestris extract with existing antibiotics.
   The absence of in vivo validation limits the practical application of findings.
   The potential variability in extract composition due to environmental or seasonal factors is not discussed.

The figures and tables are well-organized, providing clear data visualization. The chromatograms and inhibition assays effectively support the findings. However, improving figure captions with more descriptive explanations would enhance clarity.

The paper is written in clear and professional English, with minor grammatical inconsistencies. Proofreading for minor typographical errors and improving flow in certain sections (e.g., results discussion) would enhance overall readability.

Author Response

The title, "Phytochemical Composition and Antimicrobial Properties of Macleania rupestris Fruit Extract: Insights into Its Antibacterial, Antibiofilm, and Antifungal Activity", accurately describes the paper’s content. It is specific and informative, clearly indicating the study's focus on the phytochemical composition and antimicrobial effects of Macleania rupestris fruit extract.

The introduction provides a well-rounded overview of Macleania rupestris, covering its traditional medicinal uses, phytochemical composition, and the relevance of studying its antimicrobial properties. The background on antibiotic resistance and the need for alternative natural compounds is well-integrated. However, while the introduction references prior research on related species, it could provide a more in-depth discussion of previous studies specifically on M. rupestris fruit.

Response:

We appreciate the reviewer's suggestion. We have expanded our description of this plant's traditional therapeutic applications. However, we found limited references regarding research on the biological activity of this plant's extracts or studies evaluating its fruit extracts' biological activity. Consequently, we cannot conduct a more thorough examination of the existing research. 

The research design appears appropriate for evaluating the antimicrobial potential of M. rupestris fruit extract. The methods for phytochemical characterization, antibacterial and antifungal assays, and biofilm inhibition tests are described in detail. The use of standard protocols, such as the microdilution method and RRLC liquid chromatography, strengthens the study’s credibility. However, additional clarification on controls used in antimicrobial assays and potential extraction limitations would improve transparency. 

Response: We appreciate this observation and have disclosed more details about our methods of phytochemical characterisation (Lines 82-96). Information about controls have been updated (Line 221-223)

The results are presented in a clear and structured manner, with data tables and figures effectively illustrating findings. The conclusions are well-supported by the results, emphasizing the selective antibacterial activity of the extract and its biofilm inhibition potential. The discussion effectively places findings in the context of existing literature, explaining potential mechanisms of action. However, the lack of antifungal activity is briefly addressed and could benefit from a more detailed discussion.

Response: We thank the reviewer for their constructive feedback and for recognizing the clarity of our results and discussion. As suggested, we have expanded the discussion on the lack of antifungal activity by further contextualizing our findings with additional references and comparative insights. While our original discussion already addressed the structural complexity of fungal cell walls and the possible limitations of the extract’s hydrophilic compounds in interacting with fungal membranes, we have now elaborated on the metabolic adaptation of Candida species – Expanding on how Candida’s ability to utilize certain organic compounds as nutrients might contribute to the observed lack of inhibition.

The cited references are relevant and up-to-date, covering phytochemistry, antimicrobial mechanisms, and prior research on natural antibacterial compounds. There is a good balance of primary research articles and reviews. However, a more extensive discussion of studies related to M. rupestris fruit extract would strengthen the literature review. 

Response:

We agree with the reviewer; however, as mentioned in the introduction, we have not found research on M. rupestris fruit extracts, which limits the discussion on this specific issue.

This study provides a valuable contribution by identifying potential antibacterial and antibiofilm properties of M. rupestris, particularly against drug-resistant bacteria. The paper’s novelty lies in its focus on the fruit extract rather than the leaves, which have been more extensively studied. Further studies on compound fractionation and mechanisms of action would enhance the contribution. 

Response:

We appreciate the reviewer's comment and have expanded the conclusion to emphasize this important point.

The discussion does not explicitly address study limitations, which should be included. Potential limitations include:

   The study does not explore the synergistic effects of M. rupestris extract with existing antibiotics.

   The absence of in vivo validation limits the practical application of findings.

   The potential variability in extract composition due to environmental or seasonal factors is not discussed.

Response: We appreciate this observation and have incorporated more information (Lines 294 -296)

The figures and tables are well-organized, providing clear data visualization. The chromatograms and inhibition assays effectively support the findings. However, improving figure captions with more descriptive explanations would enhance clarity.

Response: We appreciate this observation and have incorporated more information (Lines 316-317)

Reviewer 2 Report

The study focuses on the phytochemical composition and bioactive activity of the ethanolic extract of M. rupestris fruit. The manuscript is well presented and justified. A wide range of complementary analyses are present, although those related to antioxidant activity are not included.

It is well known that the composition and concentration of bioactive compounds would depend on several biological factors, such as the season or the maturity degree of the substrate taken. Apparently, this has not been taken into account in the study and I think it is an important drawback.

Abstract

Lines 26-27: Indicate ranges of bioactive compound concentrations.

Keywords

Include: In vitro assays.

Introduction

Lines 49-50: Provide more information on such previous studies.

Line 52: And with antioxidant activity ?

Material and methods

Lines 74-76: Provide more information on the collection procedure of samples. Season, why such season, total weight, sampling procedure, different batches, …

Season and maturity degree are factors that may influence the concentration of the molecules to be measured.

Line 83: Indicate the storage temperature.

Line 203: Indicate the number of replicates carried out, i.e., the number of independent trials carried out.

Results

Tables 1 and 2: Indicate the number of replicates. Also in others.

Conclusions

The authors ought to express what is really new in this study in comparison to previous related research on the concentration of this berry.

No indication is provided regarding the effect of a wide range of factors on the concentration of the mentioned bioactive compounds. None of such factors have been taken into account in this study.

Author Response

The study focuses on the phytochemical composition and bioactive activity of the ethanolic extract of M. rupestris fruit. The manuscript is well presented and justified. A wide range of complementary analyses are present, although those related to antioxidant activity are not included.

It is well known that the composition and concentration of bioactive compounds would depend on several biological factors, such as the season or the maturity degree of the substrate taken. Apparently, this has not been taken into account in the study and I think it is an important drawback.

Response: We appreciate the reviewer´s comment. We incorporate any changes made. Additionally, pictures of the fruit that can showcase the level of maturity can be found in Supplementary materials Figure S1

Detail comments

Abstract

Lines 26-27: Indicate ranges of bioactive compound concentrations. 

 Response: We appreciate the reviewer´s comments. We incorporate the suggested information. 

Keywords

Include: In vitro assays. 

Response: Thank you for the suggestion. We have included in vitro assay in the keyword list.

 Introduction

Lines 49-50: Provide more information on such previous studies. 

Response:

We appreciate the reviewer's suggestion, and as suggested, we have expanded our description of this plant's traditional therapeutic applications.  

Line 52: And with antioxidant activity ? 

 Response:

Antioxidant activity was added section 2.9

Material and methods

Lines 74-76: Provide more information on the collection procedure of samples. Season, why such season, total weight, sampling procedure, different batches

Response: We appreciate this observation and agree that more details needed to be provided. We have made these changes in Lines 82-99

Season and maturity degree are factors that may influence the concentration of the molecules to be measured.

Line 83: Indicate the storage temperature.

Response: We have made these changes in Line 85-87

Line 203: Indicate the number of replicates carried out, i.e., the number of independent trials carried out. 

 Response: We have indicated the number of replicates in Lines 132, 145, 158, 223-224

Results

Tables 1 and 2: Indicate the number of replicates. Also in others. 

 Response: Details have been disclosed in line 287 and line 314

Conclusions

The authors ought to express what is really new in this study in comparison to previous related research on the concentration of this berry. 

Response:

We have expanded the conclusion to emphasize this important point.

No indication is provided regarding the effect of a wide range of factors on the concentration of the mentioned bioactive compounds. None of such factors have been taken into account in this study. 

Response: We appreciate all the reviewer's comments and we have mentioned the impact of other factors on the concentration of bioactive compound throughout the entire article 

Round 2

Reviewer 2 Report

The manuscript has been performed according to previous comments and criticisms. I would recommend its acceptation.

The manuscript has been performed according to previous comments and criticisms. I would recommend its acceptation.